# Obesity, Bone Loss, and Periodontitis: The Interlink

**DOI:** 10.3390/biom12070865

**Published:** 2022-06-22

**Authors:** Pengfei Zhao, Aimin Xu, Wai Keung Leung

**Affiliations:** 1Faculty of Dentistry, The University of Hong Kong, Hong Kong SAR, China; pfzhao@connect.hku.hk; 2Department of Medicine, Li Ka Shing Faculty of Medicine, The University of Hong Kong, Hong Kong SAR, China; amxu@hku.hk

**Keywords:** alveolar bone loss, bone and bones, bone remodeling, obesity, periodontitis

## Abstract

Obesity and periodontitis are both common health concerns that have given rise to considerable economic and societal burden worldwide. There are established negative relationships between bone metabolism and obesity, obesity and diabetes mellitus (DM), and DM and periodontitis, to name a few, with osteoporosis being considered a long-term complication of obesity. In the oral cavity, bone metabolic disorders primarily display as increased risks for periodontitis and alveolar bone loss. Obesity-driven alveolar bone loss and mandibular osteoporosis have been observed in animal models without inoculation of periodontopathogens. Clinical reports have also indicated a possible association between obesity and periodontitis. This review systematically summarizes the clinical periodontium changes, including alveolar bone loss in obese individuals. Relevant laboratory-based reports focusing on biological interlinks in obesity-associated bone remodeling via processes like hyperinflammation, immune dysregulation, and microbial dysbiosis, were reviewed. We also discuss the potential mechanism underlying obesity-enhanced alveolar bone loss from both the systemic and periodontal perspectives, focusing on delineating the practical considerations for managing periodontal disease in obese patients.

## 1. Introduction

Obesity is defined as someone with a body mass index of >30 kg/m^2^ [1] with excessive body fat accumulation leading to exaggerated health risks. According to the World Health Organization, the prevalence of obesity is increasing worldwide. For instance, it tripled from 1975 to 2016, with nearly 13% of adults (11% men and 15% women) being considered as obese [2]. The pathogenesis of multiple diseases, including type 2 diabetes mellitus (T2DM), cardiovascular disease, cancer, and osteoporosis, is associated with obesity. Osteoporosis manifests as impaired bone quality and increased risk of fracture, resulting in a reduction in mobility and quality of life [3]. The majority of clinical studies have suggested a correlation between obesity and reduced bone mass [4,5]. The systemic bone loss caused by obesity is expressed as various severities of osteoporosis and could involve alveolar bone resorption. According to the 2017 periodontal diseases classification, obesity was recognized as a significant metabolic disorder that is associated with loss of periodontal tissues [6], and an increased risk of periodontitis in obese individuals, suggesting a comorbidity effect between obesity and periodontitis. In particular, inflammatory environment and imbalanced bone homeostasis caused by obesity were widely recognized [7]. A series of studies have explored the possible molecular links between obesity and periodontitis, highlighting the concepts of shared inflammatory pathways and immune dysfunction [8,9]. However, these reviews focused on the connections between obesity and periodontal disease from the perspective of bone, rather than obesity and its negative systemic influences. With the increasing trends of obesity worldwide, there is currently insufficient evidence to guide clinicians on how to effectively manage both health problems in obese individuals with periodontitis. Therefore, this review aims to provide a summary of current epidemiologic evidence and possible biological interlinks concerning obesity. This includes its negative systemic influences—particularly bone homeostasis, and hence increased risks for periodontitis, followed by a discussion on practical considerations for the periodontal management of obese patients.

## 2. Obesity and Periodontitis: Epidemiological and Clinical Association

The first report to suggest an association between obesity and periodontitis was the 241 Japanese cross-sectional study by Saito et al. [10]. More studies, by other researchers, were later conducted, focusing on obesity as a risk indicator for periodontitis. Al-Zahrani et al. [11] extracted data from the third National Health and Nutrition Examination Survey (NHANES III) and found a significant association between obesity and the prevalence of periodontitis in young adults with an adjusted odds ratio (OR) of 1.37. Since then, many systematic reviews and meta-analyses have been published. We performed a search in the PubMed, EMBASE and Web of Science databases and identified 11 systematic reviews concerning this important association (search strategy summarized in Appendix A).

In summary, eight reports were a systematic review with meta-analysis [12,13,14,15,16,17,18,19], while three studies only attempted a systematic review [20,21,22]. Relevant characteristics and findings are summarized in Table 1. These studies were published between 2010 and 2022. One 2010 report [12] designed its own scale to evaluate the quality of the studies reported. Nevertheless, 4/1/1/1 studies used the Newcastle-Ottawa Quality Assessment Scale (N-OQAS)/Downs and Black checklist/the Critical Appraisal Checklist/STROBE checklist, while three investigations [14,20,21] reported no evaluation of quality concerning the included studies. One of the three systematic reviews [22] claimed to use N-OQAS, but without any account of the outcomes. Two reports focused on the systematic review and meta-analyses of children, adolescents, or young adults [15,18] (589 or 1983 participants) while another report was a systematic review concerning young adults [22]. A recent report focused on the association between obesity and periodontitis in pregnant women [19]. The assessment of risk of bias in the included papers was performed with the Risk of Bias in Systematic Reviews (ROBIS) tool [23]. Details of the ROBIS results are shown in Appendix A. Included were three systematic reviews with meta-analysis [12,14,18] that exhibited a high risk of bias due to considerable heterogeneity, a high risk of bias in primary studies included, and/or loss of quality assessment. Another three systematic reviews [20,21,22] were also rated as having a high risk of bias, mainly due to loss of quality assessment and merely descriptive results. There were five reports [13,15,16,17,19] with acceptable merits which included 5–19 primary studies for meta-analyses. The number of subjects ranged from 589–42,430, and various ethnicities were reported including Americans (mixed ethnicity), Europeans (predominantly Caucasians), Chinese (east Asians), and so on. One study focused on children and adolescence [15] while another focused on pregnant women [19] (Table 1).

Collectively, there is evidence to support the conclusion that there is a higher prevalence in some form or another of periodontal disease experience, e.g., clinical attachment level (CAL), bleeding on probing (BOP), or subgingival calculus in young individuals, or its surrogate, i.e., tooth loss, or edentulism in overweight or obese individuals. Overweight or obese pregnant women were more prone to experience periodontitis [19]. Notably, two meta-analyses conducting a subgroup analysis based on BMI values suggested an increasing risk of periodontitis with increased BMI values, inferring a dose-response relationship [13,16]. However, it is essential to be aware that BMI values and thresholds originating from epidemiological reports are often not directly applicable in individual clinical situations. Furthermore, the differences between general obesity and abdominal obesity may contribute to the uncertainty of dose-response relationship. One study compared the effect of abdominal obesity and general obesity on periodontal parameters, and found that the former was related to attachment loss and bleeding on probing, but not the latter [28].

The clinical characteristics of periodontium in obese individuals was also denoted in these reviews. Chaffee et al. [12] reported that obese individuals were prone to showing higher clinical attachment loss compared to none-obese individuals, suggesting an increased risk for the progression of alveolar bone loss. Additionally, the presence of subgingival calculus, visible Plaque Index, bleeding on probing, and PPD > 4 mm were also shown to be higher in obese children and adolescents [15]. A more recent study found that a higher consumption of sweets (candy), combined with poor hygiene habits and increased rates of gum inflammation, worsened periodontal tissue status (PPD, CAL) in obese adults [29]. Therefore, gingival and periodontal health and unsatisfactory oral hygiene status appeared to manifested more often in obese children, adolescents, and adults. In turn, obese individuals were observed to anticipate higher odds of any tooth loss or edentulism [17]. However, the evidence is not robust enough due to the heterogeneity of study designs and uncontrolled confounding variables.

The association between obesity and periodontitis in different ethnic and age groups is reasonably well demonstrated in the aforementioned studies. However, the causal—effect relationship and the direction of this causality have not been clarified, and potential biological/pathological mechanisms have not been evaluated or determined. Based on the current evidence, it can be speculated that the connection between both diseases is of a casual nature, and that this relationship would be bidirectional. Further laboratory-based and longitudinal population/cohort studies are warranted to establish this bidirectional causal relationship. Several potential pathophysiological mechanisms exploring the interactions between the current appreciations of how adiposity negatively affects the homeostasis of marrow, bone, gastrointestinal system, and beyond—thereby influencing periodontal health—will be discussed in the following sections.

## 3. How Does Obesity Influence Bone Quality?

Patients with obesity are also at a higher risk of metabolic complications including hypertension, T2DM, heart disease, non-alcoholic fatty liver, kidney disease, polycystic ovary syndrome, and cancer [30]. Over the past decade, a growing number of studies have indicated that obesity is associated with osteopenia and osteoporosis, suggesting a negative impact of obesity on bone quality, including the jawbone [31].

### 3.1. Obesity-Drives Disruptive Bone Homeostasis

Bone is a dynamic organ that is permanently in a process of resorption followed by remodeling/reconstruction as its major biological process throughout life [32]. Bone homeostasis involves a balance between bone formation and resorption, which are engineered by osteoblasts and osteoclasts [33]. The very first belief was that obesity correlated with increased bone mass, and long-term weight-bearing is beneficial for bone formation [34]. However, more recent studies have revealed links between excess fat and osteoporosis, making bone more fragile [35,36]. In some cohorts, the percentage of total fat mass is strongly and inversely associated with bone mineral density (BMD) and total bone mineral content [37,38]. Obesity exerts a detrimental effect on bone homeostasis by impairing the balance of osteoblast and osteoclast activities and increasing bone resorption. Reasons for this effect could be that excess fat in bone marrow, with an enlarged adipocyte number and size, leads to altered bone marrow stem cells (BMSCs) differentiation with decreased osteoblasts, thus harming bone integrity [7,39]. However, some studies have suggested that this effect is probably more complex and site-dependent, with a higher risk for only certain types of bone to fracture [4].

### 3.2. Systemic Bone Loss

Early studies support the favorable role of obesity in regulating BMD and bone mass due to the positive effect of weight-bearing exercise on bone quality [40]. Previous longitudinal studies have observed that BMD changes at some sites were positively related to the changes of fat mass [41]. By contrast, a multinational survey in 2011, for the first time, reported a higher prevalence of bone fragility fractures in postmenopausal women with obesity [42]. Of interest, Liu and colleagues enrolled 471 women in a cross-sectional study; after categorizing them into lean, overweight, and obese groups, the researchers observed that body fat < 33% showed positive association with bone density, while those with body fat > 33% showed the opposite effect for most of the skeletal sites that followed [43]. Despite the conflicting positive and negative actions of obesity on bone mass, which may be explained by heterogeneity in population and measurements, the current common view favors the idea that obesity is associated with poor bone quality [3]. Ample evidence from animal studies also implies the adverse effects of obesity on bone mass. The high fat diet (HFD) obesity model is most commonly used in studying the influence of adiposity [44]. Tencerova et al. [45] observed significant and consistent trabecular and cortical bone loss in 12-week HFD-fed mice, alongside bone marrow adipose tissue expansion, suggesting that bone mass reduction could be a biological consequence of coherent adiposity mechanisms. Another study showed critical deterioration in trabecular bone micro-architecture at the early stage of HFD-fed mice, leading to decreased trabecular bone density [46]. In the murine spines, HFD also resulted in significantly less trabecular bone volume in the lumbar vertebrae [47].

### 3.3. Periodontal Bone Loss

Compared to studies focusing on systemic bone, only a few reports have investigated the role of obesity on alveolar bone. As described in Section 2, obesity appeared to be an established risk indicator for periodontitis (Table 1). Animal studies have consistently shown that increased body weight could exacerbate alveolar bone loss in periodontitis [48]. One study used a long-term cafeteria (high-fat/high-carbohydrate) diet in Wistar rats, and reported that obesity and hyperlipidemia could further increase alveolar bone loss in ligature-induced periodontitis [49]. Muluke et al. [50] found that a palmitic acid (PA)-enriched HFD contributed to more bone loss in *Porphyromonas gingivalis*-induced periodontitis, and together with increased free fatty acids in serum, suggested a possible (but yet unclarified) role of PA promoting alveolar bone loss. Li et al. [51] also described increased periodontal inflammatory response and bone loss, which were both observed after local injection of bacterial lipopolysaccharides in the jaws of obese animals fed on a HFD, compared to a normal control diet.

Of interest, it was reported that alveolar bone density loss could be triggered by obesity without the necessity of periodontitis induction, (i.e., pathogenic microorganism inoculation or ligature) in both mature and growing animals to facilitate the development of periodontal disease. A previous investigation showed that an increase in weight gain can result in decreased alveolar bone crest height, suggesting obesity as a putative risk factor, even in a clinically healthy periodontium [52]. In the development of diet-induced obesity, another animal study showed impaired trabecular bone architecture and retarded periosteal bone formation in the early stage of HFD treatment (4 weeks), followed by a decrease in cortical bone density in the alveolar bone area with increased serum leptin levels [53]. All of these findings suggest that systemic alternations induced by obesity could have an impact on alveolar bone integrity.

## 4. Mechanisms Underpinning Dysregulated Bone Homeostasis in General under Obesity

Growing evidence has indicated the complex yet intricate mechanisms underpinning obesity-related bone dysregulation [54]. These involve various factors such as hyperinflammation, bone turnover, genetics, microbial dysbiosis, hypermetabolism, and local changes in the bone marrow environment (Figure 1).

### 4.1. Adiposity Associated Hyperinflammation

Adipose tissue, which plays a vital role in the pathology of obesity, is not only involved in storing energy and body composition, but is also an active endocrine organ [55]. In the development of obesity, abnormal and excessive adipose tissue exhibits significant adipokines and cytokine secretion alterations (Figure 1, higher panel) [56]. In particular, leptin and adiponectin are the most common adipokines, which act directly on bone cells including osteoblasts, osteoclasts, and BMSCs [57]. Despite increased levels of leptin being secreted by white adipose tissue (WAT) in obesity, dual actions of this adipokine on bone is observed. The positive impact was revealed by consistent evidence that leptin can lead to the proliferation of BMSCs, differentiation to osteoblasts, and the formation of mineralized nodules; however, leptin reduces the relative differentiation of BMSCs to adipocytes in vitro [58,59,60]. Animal models have also shown that knockout of the leptin receptor gene resulted in decreased femur bone volume and BMD [61]. Notably, the administration of leptin in leptin-deficiency mice caused a significant improvement in systemic bone volume and femur BMD [62]. It seems that the negative effect of leptin also plays a critical part in obesity. Ducy et al. [63] first reported that leptin inhibits bone formation via the sympathetic nervous system. Later, other studies demonstrated that leptin binds to its hypothalamic receptors on the gold thioglucose-sensitive neurons, subsequently activating β2 adrenergic receptors (Adrb2) in osteoblasts, downregulating the expression of *c-myc* and increasing the production of cyclin D, ultimately reducing osteoblast proliferation. The activation of Adrb2 also promotes receptor activator of nuclear factor-kappa B ligand (RANKL) expression via protein kinase A-activating transcription factor 4 pathway, and in turn, enhances bone resorption of osteoclasts [64,65]. A mouse model of neuron cell-specific deletion of the leptin receptor led to increased bone formation and trabecular volume in the spine and distal tibia [66]. Moreover, leptin, accompanied by the production of serotonin in the hypothalamic neurons, also causes decreased bone formation [67]. To summarize, these results suggest that the role of leptin in regulating bone in obesity remains controversial.

Adiponectin—a fat hormone also secreted by WAT—has been verified to be downregulated in obesity. This condition presents a negative effect on osteoblastogenesis, while increasing osteoclastogenesis due to various properties of adiponectin [68]. It has been reported that human adiponectin has the capacity to induce osteogenic differentiation of BMSC through P38 MAPK pathway, which enhances the expression of osteogenic proteins including cyclooxygenase 2 and bone morphogenetic protein 2 [69,70]. Similarly, the activity of alkaline phosphatase (ALP), formation of type I collagen and osteocalcin, and mineralized substrates can be upregulated by adiponectin in a dose-dependent manner, suggesting an effect on osteoblasts maturation [71]. Additionally, adiponectin also attenuates the proliferation and survival of osteoclast precursor cells, and leads to the decline of osteoclast regulators such as acid tartrate-resistant phosphatase and cathepsin K [72].

The obesity state is associated with the hyperplasia and hypertrophy of adipocytes, which causes chronic, low-grade systemic inflammation [73]. Inflammatory cytokines secreted by adipose tissue have been shown to increase bone resorption. An increased level of tumor necrosis factor alpha (TNF-α) initiates the link between obesity and inflammation [74]. TNF-α inhibits osteoblastogenesis through binding tumor necrosis factor receptor 1 and activating the nuclear factor kappa-light-chain-enhancer of activated B cells, extracellular signal-regulated kinases 1/2, and c-Jun N-terminal kinase pathways [75,76]. Moreover, TNF-α favors osteoclast formation and activities by different mechanisms including NF-κB/Fos proto-oncogene/nuclear factor of activated T-cells pathway independently of RANKL levels, and also stimulates the expression of RANKL and colony-stimulating factor 1 (CSF1) in stromal cells [77]. Similarly, other inflammatory cytokines such as interleukin (IL)-1, monocyte chemotactic protein-1 (MCP-1), tumor necrosis factor (TNF)-related apoptosis-inducing ligand, RANKL, IL-6, and TNF superfamily member 14 from adipose tissue are also involved in activating the formation of bone-resorbing osteoclasts, as reviewed in recent publications [78,79].

### 4.2. Bone Marrow Adiposity-Related Bone Turnover

Marrow adipose tissue (MAT), collectively consisting of bone marrow adipocytes, is an essential fat depot and accounts for nearly 8% of total fat mass storage in adults [80]. Animal and human studies have shown that MAT expands due to diseases such as obesity, osteoporosis, and diabetes [81]. It is widely recognized that obesity associated bone loss was exhibited via a greater amount of bone marrow fat fraction (Figure 1, higher panel). Wehrli et al. [82] indicated that bone marrow adipose tissue in the spine could be an explicit risk factor for bone fractures in humans. Adipocytes in bone marrow are in close contact with bone cells and hematopoietic cells. Bone marrow adipocytes were responsible for the secretion of adipokines, and some of the inflammatory cytokines mentioned above [73]. Importantly, adipocytes and osteoblasts share the same embryogenic origin, which is the pluripotential, BMSC. BMSCs are destined to be differentiated into osteoblasts or adipocytes [83]. The literature reported that transcriptional activator peroxisome proliferator-activated receptor gamma could direct the differentiation toward adipocytes in obesity [84]. The Wnt/β-catenin signaling pathway can drive BMSC toward osteoblast differentiation, which is inhibited in obesity. Secreted frizzled-related protein 1, an inhibitor of Wnt/β-catenin signaling, has been reported to be increased in mild obesity, resulting in increased MAT formation [85].

Obesity also drives a shift in the differentiation of hematopoietic stem cells (HSCs) and the disruption of hematopoiesis in the bone marrow niche, which is modulated by transcription factors such as growth factor independent 1 transcriptional repressor, Pu.1, and GATA binding protein 3 [86,87,88]. These transcription factors, upregulated by obesity, can contribute to diminished lymphocyte populations, increased numbers of myeloid progenitors, compromised immune function, and subsequent suppression of HSCs proliferation [89,90]. For example, one animal study used the HFD model to accelerate age-related trabecular bone loss and markedly reduce B-cell population in bone marrow, together with a damaged bone marrow niche resulting in reduced bone quality [91]. This immune dysfunction accounts for altered cytokines and systemic inflammation, as described previously.

### 4.3. Genetic Factors

Bone and fat metabolism share common progenitor cells, namely mesenchymal stem cells. A variety of physical, chemical, and biological factors lead to disruption in mesenchymal stem cell differentiation [31]. Various signaling pathways contribute to adipogenic and osteogenic differentiation, such as Wnt/β-catenin, Notch, and phosphatidylinositol 3-kinase-protein kinase B pathways [92]. Therefore, pleiotropic genes regulating these pathways may explain the association of obesity and osteoporosis (Figure 1, higher panel). Human genome studies indicate a strong association between human fat mass and obesity-associated (*FTO*) gene variants and BMI in different populations [93,94,95]. The *FTO* of 410.5 kb locates on chromosome 16q12.2, encoding 2-oxoglutarate (2-OG) iron (II)-dependent alpha-ketoglutarate-dependent hydroxylase family dioxygenase [96] with well-described variants linked to obesity and an increased risk of osteoporosis [97]. The FTO protein is an RNA demethylase and enables the stability of mRNA-encoding proteins which prevent osteoblasts from genotoxic damage and apoptosis [98]. An in vivo study using a Cre-lox recombination system to deplete *FTO* in osteoblasts of C57BL/6 mice revealed an increase in osteoblast death and bone loss. Furthermore, more severe bone loss and greater damage to osteoblasts were observed in such mice fed with HFD, implying that individuals carrying the *FTO* gene mutation are more prone to osteoporosis due to the increased risk of osteoblasts’ apoptosis [99]. A more recent clinical study further corroborated the obesity-related polymorphisms in the *FTO* gene with increased susceptibility to osteoporotic bone fractures [100].

With the evolution of high throughput and genomics technology, emerging evidence has uncovered several gene components that could be linked with obesity phenotypes and osteoporosis. For example, four single-nucleotide polymorphisms (SNPs) in the sprouty RTK signaling antagonist 1, or *SPRY1* gene, were significantly associated with obesity-related traits and osteoporosis [101]. Another cross-sectional study revealed that the minor alleles A/C of rs7117858 SNP, located downstream of SRY-box transcription factor 6 or *SOX6*, was associated with lower bone mass, as measured by quantitative ultrasonography and obesity in Caucasian young adults [102]. However, the molecular mechanism underlying the gene variants associated with bodily changes remains unclear.

### 4.4. Gut Microbiota

Previous studies have identified an association between gut microbiota dysbiosis and multiple diseases, e.g., T2DM, hypertension, Alzheimer’s disease, and obesity. Growing evidence supports the role of gut microbiota in regulating bone health in obesity (Figure 1, higher panel) [103]. Collins et al. (2015) reported that HFD-fed rats developed greater severity of osteoarthritis than their lean counterparts and gut microbiota dysbiosis in obese rats showed a strong predictive relationship with metabolic osteoarthritis [104]. Another report in the same year demonstrated that structural changes of gut microbiota were associated with an altered bone niche in obesity, leading to impaired HSC differentiation and increased bone loss [105]. A more detailed discussion on this topic can be found in Fernández-Murga et al. [106]. They identified a specific intestinal bacterium, *Bifidobacterium pseudocatenulatum* CECT 7765, and proved its role in regulating bone health. In HFD-fed mice, reduced BMD in the trabecular bone and deteriorated trabecular architecture were detected in the femur. Oral supplementation with *B. pseudocatenulatum* CECT 7765 in obese mice attenuated the negative effect on bone microstructural changes, together with upregulated Wnt/β-catenin pathway gene expression.

Although the crucial role of gut microbiota can be inferred from numerous studies, the key mechanism underlying the association remains not fully elucidated, hence further studies are warranted. Following the principal findings on gut microbiota, probiotics, prebiotics, and microbial transplantation could be potential treatment protocols for osteoporosis in obesity via manipulation of gastrointestinal microbiota, hence its related components and metabolites.

### 4.5. Other Factors: Diet and Hypermetabolism

There is also compelling evidence showing that diet itself may be associated with obesity-related bone metabolism, particularly fatty acid content (Figure 1, higher panel). An animal study focusing on the links between obesity and osteoarthritis used different high-fat diets composed of saturated fatty acids (SFAs), ω-6 polyunsaturated fatty acids (PUFAs), and ω-3 PUFAs. The results show that SFAs and ω-6 PUFAs independently worsen injury-induced knee osteoarthritis and bone resorption [107]. Octanoic acid, a major component of ketogenic dietary supplements, was also found to be detrimental to the trabecular bone microarchitecture of the femur and tibia as well as bone ALP levels [108]. It has been suggested that SFA and ω-6 PUFA intake promotes low-grade chronic inflammation of macrophages and regulates mesenchymal stem cell lineage commitment [109].

Another possible mechanism could be obesity-induced hypermetabolism resulting in accelerated senescence in bone cells. Tencerova et al. [110] investigated 54 individuals and divided them into lean, overweight, and obese groups based on their BMI values. BMSCs were then extracted and RNA sequencing performed. The results showed an upregulated expression of metabolic genes involved in glycolytic and oxidoreductase activities. Moreover, insulin receptor-positive and leptin receptor-positive cells were increased in BMSC cultures, which were prone to differentiate into adipocytes. It has been reported that adipocyte-like cells prefer mitochondrial oxidative phosphorylation and ROS production compared to osteoblast-like cells, which may drive BMSC senescence [111]. Ultimately, BMSCs from men with obesity expressed with increased ROS and higher senescent markers, suggesting a possible role of senescent cells in poor bone formation.

## 5. Mechanisms by Which Obesity Enhances Periodontal Bone Loss

Alveolar bone is essential for tooth support, and obesity-related alveolar bone loss can be attributed to systemic changes, as mentioned above, as well as several potential mechanisms that happen at the periodontium. This section aims to summarize all possible mechanistic links concerning this phenomenon (Figure 1, lower panel).

### 5.1. Hyperinflammation in Periodontium

In response to external and internal stimuli, immune cells in the periodontium, such as macrophages and lymphocytes, release multiple proinflammatory cytokines, including ILs, TNF-α, and matrix metalloproteinases (MMPs) [112]. This inflammatory cascade contributes to osteoclastogenesis and, in turn, alveolar bone loss through increased binding of RANK and RANKL (Figure 1, lower panel) [113]. In obese subjects, the levels of proinflammatory cytokines are increased in serum and hence in gingival crevicular fluid (GCF), suggesting local hyperinflammation. A meta-analysis compared cytokine profiles in the GCF of chronic periodontitis patients with and without obesity and reported significantly higher levels of GCF TNF-α, resistin, and IL-1β in obese periodontitis patients, suggesting a higher level of localized inflammation and a driver of bone loss in the latter [114] (methodological quality: Appendix A). There is also evidence showing elevated levels of proinflammatory adipokines (leptin) and decreased anti-inflammatory adipokines (adiponectin) in the GCF of obese patients with/without periodontitis, a direct reflection of the systemic changes of proinflammatory cytokine concentration in obesity [115].

Furthermore, the imbalance of reactive oxygen species and antioxidant capacity may connect obesity and alveolar bone loss. Given that oxidative stress enhances alveolar bone loss, the level of 8-hydroxydeoxyguanosine, an oxidized nucleoside, was shown to be increased in obese rats with periodontitis, while the ratio of reduced/oxidized glutathione decreased compared to lean counterparts with periodontitis [116,117]. When periodontitis was induced in such an animal model, higher oxidative stress was observed with greater alveolar bone resorption. Human studies also revealed an increased level of periodontal oxidative stress in obese patients compared to those of normal weight. A noteworthy correlation analysis showed that the parameters of oxidative stress, such as malondialdehyde, protein carbonyl, and total antioxidant capacity in GCF, were associated with clinical attachment loss, which reflected the corresponding alveolar bone status [118]. However, further mechanistic studies on a plausible link between obesity, periodontal oxidative stress, and alveolar bone loss are still required.

### 5.2. Resident Immune Cells’ Dysregulation

There is clear evidence that immune responses to periodontal bacteria are dysregulated by obesity. Importantly, the phagocytic capacity and defective oxidative burst of macrophages were impaired in obese mice [119]. A study investigating the role of macrophages in obesity with periodontitis observed that obesity was associated with reduced periodontal tissue infiltration and activation of macrophages, thereby aggravating the progression of periodontitis [120]. However, the detailed alterations of different immune cells such as myeloid-derived cells, T cells, and B cells induced by obesity in the periodontal environment are still lacking. It is notable that macrophages and osteoclasts share the same myeloid cell lineage and common monocytic cell embryology [121]. Recently, Kwack et al. [122] postulated a conceptual “two-hit” model to explain obesity-exacerbated alveolar bone destruction, namely that obesity expands specific myeloid progenitor subpopulations (first hit), then is transported to the periodontal inflammatory site to differentiate into monocytes (osteoclasts) which led to obesity-associated alveolar bone loss (second hit) in a context-dependent manner (Figure 1, lower panel).

In 2009, Zhou et al. proposed that the homotolerance effect, defined as subsequent immune unresponsiveness when exposed to the same agonists, may participate in obesity-related periodontal destruction [123]. Free fatty acids’ increase with obesity and/or under HFD could bind to toll-like receptor (TLR) 2, similar to lipopolysaccharides (LPS), triggering development of homotolerance [124,125]. As a result, there will be an attenuated innate immune response to microbial attack, which facilitates periodontal destruction [126]. However, no further evidence supports the homotolerance theory in the periodontal context, and controversial results in recent years have shown that free fatty acids such as PA amplified LPS-induced macrophage immune response via the TLR4 signal but not the TLR2 signal, subsequently increasing IL-6, MCP-1, and CSF-1 levels [127,128]. Thus, further studies into free fatty acids and periodontal immune regulation are needed.

### 5.3. Periodontal Osteoblast and Osteoclast Activity

Both obesity and periodontal bacteria can suppress differentiation and induce apoptosis of osteoblasts, thereby deteriorating osseous coupling [129,130]. It has been reported that immune responses caused by bacterial infection contributed to osteoblastic cell loss. A study described significant apoptotic responses in New Zealand obese (NZO) mice primary calvarial osteoblast culture exposed to *P. gingivalis* but not in *P. gingivalis*-treated primary calvarial osteoblasts from control C57BL/6J mice [131]. Suppressed osteoblastic marker expression was also observed in NZO osteoblasts including alpha-1 type I collagen, bone gamma-carboxyglutamate protein, and RUNX family transcription factor 2 (Runx2). The HFD-induced model also suggested that obesity downregulated Runx2 in the periodontal area [132].

In contrast, obesity has been shown to enhance osteoclastogenesis in periodontal areas (Figure 1, lower panel). Osteoclast numbers, indicated by tartrate-resistant acid phosphatase staining, showed a twofold increase in mice fed with HFD and induced periodontitis compared to non-obese mice with periodontitis [51]. The interaction of RANKL and RANK on the surface of osteoclasts is responsible for osteoclast formation and bone-resorption activity [133]. Osteoprotegerin (OPG) can block the function of RANKL by competitively binding to RANK. Numerous studies have reported an increased ratio of RANKL/OPG in the periodontal tissue of the obese model, suggesting an activated osteoclast function in the animal [132,134]. The hyperinflammation, increased oxidative stress, and immune cell dysfunction mentioned above can interact with osteoclast formation and function, leading to bone loss. Furthermore, emerging evidence suggests the role of specific saturated fatty acids in regulating osteoclastogenesis. A PA-enriched HFD enhanced obesity-related alveolar bone loss and osteoclast formation, but not an oleic acid (OA)-enriched HFD [135]. Further evidence found that PA negatively influences human periodontal ligament fibroblasts, including increased cell death, matrix degradation markers, and RANKL/OPG ratio [136].

### 5.4. Periodontal Microbiota

The composition and diversity of periodontal microbiota can be altered by obesity (Figure 1, lower panel). Haffajee et al. [137] reported a higher proportion of *Tannerella forsythia* in the subgingival biofilms of overweight and obese individuals than in those from individuals of normal weight. Moreover, *T. forsythia*, *Fusobacterium* spp., and *P. gingivalis* were also reported to be elevated in the saliva of obese patients with or without periodontitis [138]. A further cross-sectional study reported that obese patients with periodontitis harbored higher levels of periodontopathogens than those of normal weight with periodontitis, including *Aggregatibacter actinomycetemcomitans*, *Eubacterium nodatum*, *Fusobacterium nucleatum* sub spp. *vincentii*, *Parvimonas micra*, *Prevotella intermedia*, *T. forsythia*, *Prevotella melaninogenica*, and *Treponema socranskii* [139]. However, the causal relationship between obesity-induced changes in these pathogenic microorganisms and alveolar bone loss is yet to be clarified.

Despite oral and gut microbiome compositions being distinct, some bacteria are common in the oral and gut environment since they share an intramucosal connection [140]. There may be a link between the oral and gut microbiomes which affects obesity-related periodontal destruction. A recent study using fecal microbial transplantation and a ligature-induced periodontitis model demonstrated that gut microbial dysbiosis from HFD-fed mice enhanced alveolar bone destruction and altered the composition of the periodontal microbiome of recipient mice [141]. They proposed uric acid elevation in the periodontal tissue could be the mechanism aggravating alveolar bone loss.

## 6. Prevention and Treatment of Periodontal Bone Loss in Patients with Obesity

Given that the prevalence of obesity and associated periodontitis is growing globally, it would be a frequent occurrence to encounter such patients in dental practices. To date, no census or guideline has been recommended or reported to treat periodontitis associated with obesity, other than the same protocol for non-obese periodontitis patients. A targeted and personalized management approach is required due to the multifactorial nature of obesity and periodontitis. Based on the mechanisms discussed above, several considerations can be raised for management of this unique cohort.

### 6.1. Basic Periodontal Therapy

In light of the crucial role of microbiota-associated inflammation in the development of periodontitis, basic periodontal therapy is essential for preventing periodontal bone loss. The basic process of prevention, nonsurgical periodontal treatment, and surgical intervention can be followed in obese patients. Risk-factor control, including dietary counselling and weight watching, could be important in the systemic phase and/or prevention stage in terms of addressing the association between obesity, elevated systemic and periodontal inflammation, and periodontitis. Results concerning the impacts of obesity on treatment response to nonsurgical periodontal therapy remain varied. While some reports did not observe any significant differences in the periodontal parameters, including PPD and CAL, between obese and non-obese individuals following nonsurgical periodontal treatment [142,143], others indicated an inferior response to periodontal treatment in obese patients [144,145,146] (methodological quality of [146]: Appendix A). Increased confounder control and a more rigid study design are perhaps warranted to determine the clinical effects of periodontal therapy in obese patients. However, less improvement in clinical parameters was commonly observed in obese patients. In this regard, weight loss intervention could be considered an additional benefit since serum proinflammatory cytokines tend to reduce after weight loss and the likelihood of obese individuals becoming DM would definitely diminish. A conveniently sampled clinical trial indicated that weight loss through dietary therapy improved the response of obese subjects to nonsurgical periodontal treatment [147]. Obese patients who had significant weight loss after bariatric surgery also showed a more remarkable improvement in periodontal parameters after nonsurgical treatment than those who did not have such surgery [148]. Similar to smoking cessation, it is essential to provide targeted supportive periodontal therapy, including individually-tailored nutritional counselling, an exercise routine for weight reduction, and better oral hygiene practices. According to the baseline periodontal health and overall physical status of obese patients, supportive therapy frequency should be individualized [9].

### 6.2. Combating Periodontal Bone Loss and Obesity with Exercise

Exercise is considered an effective non-pharmacological method to regulate obese-related metabolic disorders and bone health. Laboratory studies have provided evidence and information about the benefits of exercise on periodontal health, especially in obese models. In 2011, Azuma et al. [149] observed that obesity-induced gingival oxidative stress could be effectively inhibited by exercise through systemic ROS reduction, which was also observable in serum and gingival tissue. It has also been reported that exercise attenuated alveolar bone loss and anxiety-like behavior in a ligature-induced periodontitis rat model, which has drawn the attention of clinicians and researchers towards exercise and periodontitis [150]. A study on HFD and streptozotocin-induced diabetic rats with ligature-induced periodontitis concluded that eight weeks of exercise significantly reduced alveolar bone loss and improved serum inflammatory profiles compared to the sedentary control [151]. A more recent study reported that mice with physical training developed increased BMD, trabecular bone volume, and bone volume/total volume ratio, as well as an increased number of osteoblasts, as opposed to a decreased number of osteoclasts in maxillary bone in the orthodontic tooth movement process [152].

Most clinical studies support a relationship between physical exercise and attenuated periodontal destruction. An epidemiological survey comprising 12,110 participants in the US, after controlling for factors such as gender, age, and smoking, etc., provided a striking conclusion: individuals who maintained normal body weight, regular recommended levels of exercise, and a high-quality diet were 40% less likely to experience periodontitis [153]. In the systematic review conducted by Ramseier et al. [154] (methodological quality: Appendix A), two studies concerning the impact of physical exercise interventions in periodontitis patients were identified. One clinical trial compared undescribed ‘standard periodontal treatment’ with yoga exercise with ‘periodontal treatment’ alone from a cohort of conveniently sampled patients at a dental outpatient clinic [155]. The other clinical trial involved 71 conveniently recruited obese men who participated in an exercise or dietary intervention program with unknown periodontal disease conditions and an unknown treatment protocol. The same study lacked an obese man group with periodontal treatment only [156]. Both studies reported significantly improved periodontal parameters, such as BOP, PPD ≥ 4 mm, and CAL changes, after 12 weeks. Such a low-quality clinical report nevertheless suggests the beneficial effect of exercise on periodontal condition. Considering the limited evidence so far, as well as the undefined periodontal intervention reported in both studies [134,135], the effect of exercise in the systemic phase of periodontal therapy should be considered unknown.

### 6.3. Systemic or Local Adjunctive Therapy

It seems that attenuation of the host’s immune reaction to microbial plaque, thereby leading to a decrease in the ratio of RANKL/OPG and a reduction in associated bone loss, can be the desired outcome of adjunctive therapy to periodontal treatment. In fact, melatonin has been administered in obese subjects and shown a significant decrease in systemic proinflammatory cytokines and adiposity [157]. Virto et al. [158] reported that melatonin dissolved in drinking water as adjunct to standard mechanical debridement helped to reduce alveolar bone loss and proinflammatory cytokines in Wistar rats with comorbidity of periodontitis and obesity. Supplementation with probiotics has been suggested as a possible adjunct for periodontitis management. *Akkermansia muciniphila*, a gut commensal with anti-inflammatory properties, has been administered in overweight and obese patients and contributed to the regulation of several metabolic parameters [159]. An animal study further investigated the effect of oral administration of pasteurized *A. muciniphila* on *P. gingivalis*-induced periodontitis in obese mice [160]. The results demonstrated that oral gavage with *A. muciniphila* reduced alveolar bone loss and systemic inflammation biomarkers, probably by decreasing the expression of *P. gingivalis* virulence factors. However, human evidence suggesting the clinical use of adjunctive therapy to standard periodontal treatment in periodontitis patients with obesity is currently lacking. Only one randomized controlled trial supports the use of adjunctive antimicrobial photodynamic therapy to scaling and root planing in improving the clinical parameters (PPD, CAL, BOP) in obese patients with periodontitis [161]. Future therapeutic options targeting the regulation of host immune responses and microbiota dysbiosis should be considered as promising.

### 6.4. Future Perspectives

A major goal of periodontal therapy is to control microbial infection and reconstruct the tooth supporting tissue, particularly alveolar bone. Managing periodontitis patients with obesity requires a comprehensive understanding of pathophysiology, microbiology, pharmacology, and nutrition by dentists. Future advances in basic science are desirable in order to shed light on targeted therapies and potential new drugs. Detailed and distinctive mechanistic links between various inflammatory factors, oxidative stress, and immune dysfunction in obesity could help us to develop new, specific adjunctive therapies for the modulation of host immune responses in periodontitis. Further research into gut and periodontal microbial dysbiosis in obesity can provide better guidance for new treatments such as probiotics and microbial transplantation. Although diet control and exercise are considered clinical and cost-effective strategies, more research is needed to determine their efficiency in the systemic phase of periodontal therapy as well as the precise mechanisms. At the same time, additional well-designed and high-quality randomized controlled trials are required to generate evidence of adjunctive treatment in periodontitis associated with obesity.

## 7. Concluding Remarks

Although there is a lack of evidence to determine the causal—effect relationship, the association of obesity and periodontal disease is proven. The interactions between obesity, bone health, and periodontitis are complex, and alveolar bone loss is likely to be regulated by both systemic and periodontal changes in obesity. Unravelling the exact mechanism of bone loss in obesity development, along with new findings in preclinical and clinical studies, allows us to present new insights into comorbidity and perhaps make a breakthrough in the periodontal treatment of those affected. Dentists’ awareness of the risks associated with treating obese individuals and the importance of personalized approaches should be raised. Furthermore, the improvement/refinement of routine periodontal therapy for successful management of obese subjects with periodontitis remains to be fully elucidated.

## Figures and Tables

**Figure 1 biomolecules-12-00865-f001:**
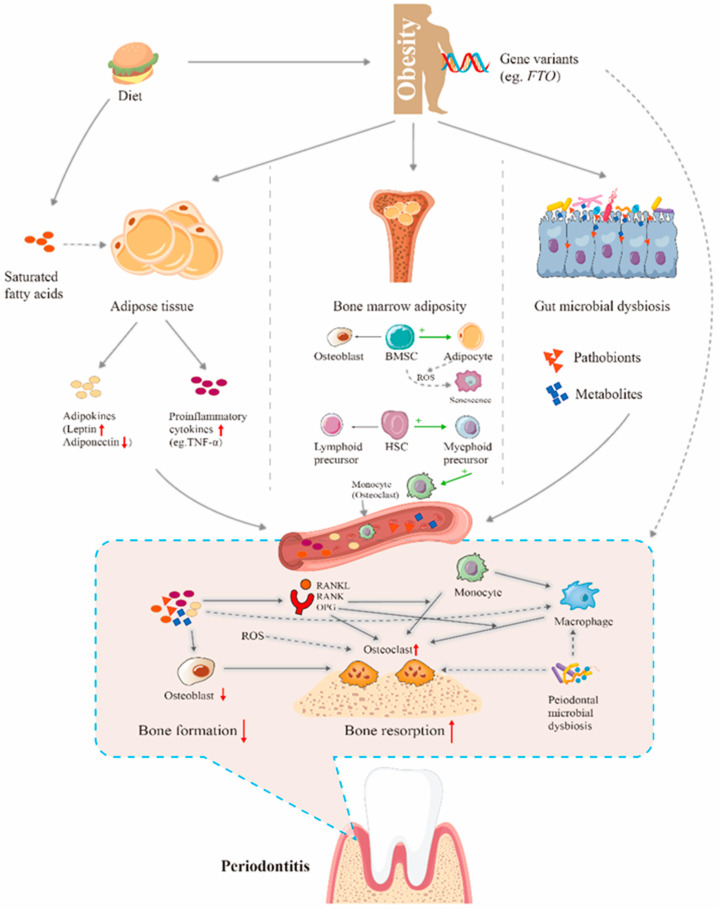
Mechanisms underpinning obesity-related systemic and alveolar bone loss. In obesity, excessive or unhealthily expanded white adipose tissue leads to the dysregulation of adipokines and inflammatory cytokines production. Apart from that, bone marrow adiposity is associated with the adipogenic differentiation of bone marrow stem cells (BMSCs) and the disruption of hematopoiesis, thereby leading to decreased osteoblasts, lymphoid precursors, and increased monocytes differentiation. Worse still, hypermetabolism in obesity can accelerate the senescence of BMSC. As obesity can trigger gut microbial dysbiosis, bone resorption inducing pathobionts, or metabolites from the gut via systemic circulation, this can translocate to bones including the human jaw. Certain gene polymorphisms could exacerbate osteoporosis in obese individuals. Additionally, diet-contained saturated fatty acid may play an independent role in promoting bone loss. These factors are involved in the disruption of bone homeostasis and could have an impact on local periodontal defense, microbial composition, or bone remodeling mechanisms including hyperinflammation, immune dysregulation, periodontal microbial dysbiosis or imbalanced osteoblast, and osteoclast activities, respectively. Red ellipsoid: saturated fatty acids; purple ellipsoid: proinflammatory cytokines; yellow ellipsoid: adipokines; red triangles: pathobionts; blue squares: metabolites. BMSC: bone marrow stem cell; *FTO*: fat mass and obesity-associated genes; HSC: hematopoietic stem cells; OPG: osteoprotegerin; RANK: receptor activator of nuclear factor-kappa B; RANKL: RANK ligand; ROS: reactive oxygen species; TNF-α: tumor necrosis factor alpha.

**Table 1 biomolecules-12-00865-t001:** Systematic reviews and meta-analyses evaluating the association between obesity and periodontal disease (in chronological order).

Authors and Year (Report Nature)	Aim and Objective	Studies Included and Disease Definition	No. of Participants (Grand Total and Range in Parenthesis) and Outcome of MA	Quality Assessment	Subgroup Analysis (I); Sensitivity Analysis (II); and Publication Bias Assessment (III)	Summary
Chaffee et al. 2010 [12] (SR and MA)	To compile the evidence concerning relationship between obesity and periodontal disease.	70/28 studies included for SR/MA. Obesity definition: BMI or WHR Periodontal disease definition—up to 18 different selected clinical criteria with top 3 as: (1) CPI = 4; (2) CPI ≥ 3; (3) ≥1 site with CAL ≥ 3 mm and PPD ≥ 4 mm.	70,855 (96–13,665) participants; ORs, or MD of CAL between obese and non-obese groups.	Using a specific scale design by the authors. 13, 7, and 8 studies were rated with high, medium, or low quality of evidence, respectively.	(I) Yes. Based on study characteristics. (II) Attempted. Exclusion of any single study only slightly altered the outcome. (III) Limited bias detected.	A positive association between periodontal disease and obesity. Overall OR: 1.35, 95% CI (1.23, 1.47). Obese patients were prone to show greater mean CAL. Summary MD = 0.58 mm; 95% CI (0.40, 0.74).
Suvan et al. 2011 [13] (SR and MA)	To systematically review the evidence investigating the association between obesity and periodontitis.	33/19 studies included for SR/MA. Obesity definition: BMI, WHR, WC, or body fat%. Periodontal disease definition—up to 16 different selected clinical criteria, with top 3 criteria are: (1) CPI ≥ 3; (2) > 4 teeth with minimum one site with CAL ≥ 3 mm and PPD ≥4 mm; (3) ≥ 1 site with CAL ≥3 mm and PD ≥ 4 mm.	39,777 (96–13,665) participants; pooled estimates of ORs.	Newcastle-Ottawa Quality Assessment Scale (N-OQAS) [24].	(I) Yes. Based on BMI categories. (II) Not reported. (III) Not reported.	Significant associations between periodontitis and obesity (OR: 1.81, 95% CI (1.42, 2.30]), or overweight (OR: 1.27, 95% CI (1.06, 1.51]), and obese and overweight combined (OR: 2.13, 95% CI (1.40, 3.26]). Results suggested a positive association of BMI categories, obese and overweight with presence of periodontitis, although the magnitude appeared unclear.
de Moura-Grec et al. 2014 [14] (SR and MA)	To systematically review the studies regarding association between overweight/ obesity and periodontitis.	31/22 studies included for SR/MA. Obesity definition: BMI or WC. Periodontal disease definition—up to 16 different selected clinical criteria, with top 3 as: (1) PPD ≥ 4 mm; (2) CPI ≥ 3; (3) PPD ≥ 5 mm.	69,089 (79–13,665) participants; ORs, MD in BMI between periodontitis and periodontally healthy group.	Not reported.	(I) Not reported. (II) Not reported. (III) Not reported.	Obesity and overweight showed an increased odds for periodontitis (OR: 1.3, 95% CI (1.25, 1.35)). Participants with periodontitis had higher BMI compared to periodontally healthy participants. MD: 2.74 kg/m2 (95% CI (2.70, 2.79]).
Keller et al. 2015 [20] (SR)	To longitudinally examine the association between obesity and periodontitis.	13 studies included for SR. Obesity definition: BMI or WC. Periodontal disease definition—7 clinical outcomes: PPD, CAL, ABL, PI, GI, CPI, BOP, and FMBS with thresholds not reported.	44,758 (46–36,910) participants; NA.	Not reported.	(I) Not reported. (II) Not reported. (III) Not reported.	Suggests overweight, obesity, weight gain, and increased waist circumference could be considered as risk factors for development of periodontitis.
Li et al. 2015 [15] (SR and MA)	To investigate the association between anthropometric measurements and periodontal diseases in children and adolescents.	16/5 studies included for SR/MA. Obesity definition: BMI or WC. Periodontal disease definition—3 clinical criteria: (1) either two sites between adjacent teeth with CAL ≥4 mm, or at least two such sites with PPD ≥ 5 mm; (2) ≥1 sites with CAL ≥ 3 mm and PPD ≥ 3 mm; (3) ≥ 1 bleeding site.	589(87–164) participants; ORs.	Strengthening the Reporting of Observational studies in epidemiology (STROBE) checklist [25].	(I) Yes, based on different periodontal markers. (II) Not reported. (III) No substantial bias detected.	Reported positive association between obesity and presence of subgingival calculus (OR: 3.07, 95% CI (1.10, 8.62]), visible Plaque Index (OR: 4.75; 95% CI (2.42, 9.34]), BOP (OR: 5.41; 95% CI (2.75, 10.63]), and risk of PPD > 4 mm (OR: 14.15; 95% CI (5.10, 39.25]) in children and adolescents. Concluded that obesity is associated with some signs of periodontal disease in children and adolescents.
Nascimento et al. 2015 [16] (SR and MA)	To systematically review the effect of weight gain on incidence of periodontitis.	Both 5 studies included for SR and MA. Obesity definition: BMI or WC. Periodontal disease and progression definition— 3 sets of clinical criteria: (1) PPD ≥ 4 mm; (2) ABL ≥ 40% or PPD or CAL ≥ 5 mm; (3) self-reported periodontal disease.	42,158 (224–36,910) participants; RRs.	N-OQAS	(I) Yes, based on obese status. (II) Attempted. Omission of any single study did not alter the findings. (III) No substantial bias detected.	Results showed overweight (RR: 1.13, 95% CI (1.06, 1.20]) and those participants who became obese (RR: 1.33, 95% CI (1.21, 1.47]) had a significant higher risk to develop periodontitis.
Nascimento et al. 2016 [17] (SR and MA)	To examine the bidirectional association of tooth loss and obesity.	25/16 studies included for SR/MA. Obesity definition: BMI. Periodontal disease and progression manifestation: number of teeth lost.	42,430 (186–16,416) participants; ORs.	The Critical Appraisal Checklist (Joanna Briggs Institute [26]).	(I) Yes, based on tooth loss or edentulism. (II) Attempted. Omission of any single study did not alter the findings. (III) Presence of a small-study effect when any tooth loss was considered as an exposure.	Results indicated obese individuals had higher odds of having any tooth loss (OR: 1.49, 95% CI (1.20, 1.86)) or being edentulous (OR: 1.25, 95% CI (1.10, 1.42]), respectively. Individual with any tooth loss had higher odds (OR: 1.41, 95% CI [1.11, 1.79]) for obesity; similar for edentulous participants (OR: 1.60, 95% CI: (1.29, 2.00)). Suggested bidirectional association between tooth loss and obesity.
Martens et al. 2017 [18] (SR and MA)	To investigate the association between overweight/ obesity and periodontal disease in children and/or adolescents.	12/7 studies included for SR/MA. Obesity definition: BMI, WHR, WC, or body fat%, and skinfold thickness. Periodontal disease definition: ≥ 1 site with CAL≥ 3 mm and PPD ≥ 3 mm.	1983 (87–1204) participants; ORs.	Downs and Black checklist [27].	(I) Not reported. (II) Attempted. Omission of any single study did not alter the findings. (III) No evidence of publication bias detected.	Significant association between periodontal disease and obesity in children (OR: 1.46, 95% CI (1.20, 1.77]).
Martinez-Herrera et al. 2017 [21] (SR)	To systematically review the association between obesity and periodontal disease.	28 studies included for SR. Obesity definition: BMI, WC, WHR, or body fat%. Periodontal disease definition—7 clinical outcomes: PPD, CAL, PI, BOP, ABL, CPI and GI with thresholds not reported.	102,221 (91–36,910) participants; NA.	Not reported.	(I) Not reported. (II) Not reported. (III) Not reported.	All studies except two articles described an association between obesity and periodontal disease.
Khan et al. 2018 [22] (SR)	To investigate if overweight or obese is risk factor for periodontitis in adolescents and young adults.	25 studies included for SR. Obesity definition: BMI, WC, WHR, or body fat%. Periodontal disease definition—up to 17 different selected clinical criteria, with top 2 as: (1) CPI ≥ 3; (2) ≥ 1 sites with PPD ≥ 4 mm.	51,597 (55–17,660) participants; NA.	N-OQAS	(I) Not reported. (II) Not reported. (III) Not reported.	Suggested evidence available indicating obesity was associated with periodontitis in adolescents and young adults.
Foratori-Junior et al. 2022 [19] (SR and MA)	To generate pooled evidence for the association between excess weight and periodontitis during pregnancy.	Both 11 studies included for SR and MA. Obesity definition: BMI. Periodontal disease definition—up to 8 different selected clinical criteria, with top 3 as: (1) ≥ 2 interproximal CAL ≥ 4 mm on different teeth; (2) ≥ 2 interproximal sites with CAL ≥ 3 mm or PPD ≥ 4 mm (on different teeth), or one site with PPD ≥ 5 mm; (3) interproximal CAL ≥ 2 on nonadjacent teeth or buccal or oral CAL ≥ 3 mm with PPD > 3 mm detectable on ≥ 2 teeth.	2152 (50–682) participants; RRs.	N-OQAS	(I) Not reported. (II) Not reported. (III) No evidence of publication bias detected.	Positive association between overweight/obesity and periodontitis during pregnancy (RR: 2.21, 95% CI (1.53, 3.17]).

ABL: alveolar bone loss; BMI: body mass index; BOP: bleeding on probing; CAL: clinical attachment level; CI: confidence interval; CPI: Community Periodontal Index; FMBS: full-mouth bleeding score; GI: Gingival Index; MA: meta-analyses; MD: mean difference; NA: not applicable; N-OQAS: Newcastle-Ottawa Quality Assessment Scale; OR: odds ratio; PI: Plaque Index; PPD: probing pocket depth; RR: relative risk; SR: systematic reviews; WC: waist circumference; WHR: waist/hip ratio.

## Data Availability

Not applicable.

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
