# Peer review of "Obesity, Bone Loss, and Periodontitis: The Interlink"

_biomolecules, 2022, doi:10.3390/biom12070865_

Round 1
Reviewer 1 Report
The review work entitled "Obesity, Bone Loss, and Periodontitis: The Interlink" provides a comprehensive summary of the clinical changes in the bones in particular periodontal tissue such as the losses to alveolar bones occurring with obesity. The work provides evidences for the mechanisms that link obesity and periodontal disease such as inflammation, immune deregulation, etc. and the practical considerations for therapy.
However, the readability and the organization of the manuscript needs to be improved.
For instance, line 138-139, "Bone is a dynamic organ, with resorption follow by remodeling/reconstruction as its major regular biological process from embryonic development to end of life" is very confusing and could have been simplified to clearly indicate the authors notion.
It may be more interesting if the authors could add an additional section suggesting the directions for future research to answer some of the key questions to improve the current therapy.
Reviewer 2 Report
It is a well-done paper, but table S1 and S2 cannot be checked due to a wrong access provided.
The research provide a summary of current epidemiologic evidence and possible biological interlinks concerning obesity, its influences in bone homeostasis, and increased risks for periodontitis. Also, discussed the practical considerations for periodontal management of obese patients.
I consider relevant and interesting. The topic provide a new item to relate periondititis with obesity through bone physiology. This item is a new way of deal with peridontitis treatment
It is well written and easy to read. The data provide clear conclusions.
They address with the main question: relationship between obesity and periodontitis dealing with bone physiology.
